# The HisCl1 histamine receptor acts in photoreceptors to synchronize *Drosophila* behavioral rhythms with light-dark cycles

Faredin Alejevski[1], Alexandra Saint-Charles[1,2], Christine Michard-Vanhée[1], Béatrice Martin[1], Sonya Galant[1], Daniel Vasiliauskas[1] & François Rouyer [1]

In *Drosophila*, the clock that controls rest-activity rhythms synchronizes with light-dark cycles through either the blue-light sensitive cryptochrome (Cry) located in most clock neurons, or rhodopsin-expressing histaminergic photoreceptors. Here we show that, in the absence of Cry, each of the two histamine receptors Ort and HisCl1 contribute to entrain the clock whereas no entrainment occurs in the absence of the two receptors. In contrast to Ort, HisCl1 does not restore entrainment when expressed in the optic lobe interneurons. Indeed, HisCl1 is expressed in wild-type photoreceptors and entrainment is strongly impaired in flies with photoreceptors mutant for HisCl1. Rescuing HisCl1 expression in the Rh6-expressing photoreceptors restores entrainment but it does not in other photoreceptors, which send histaminergic inputs to Rh6-expressing photoreceptors. Our results thus show that Rh6-expressing neurons contribute to circadian entrainment as both photoreceptors and interneurons, recalling the dual function of melanopsin-expressing ganglion cells in the mammalian retina.

[1] Institut des Neurosciences Paris-Saclay, Univ. Paris Sud, CNRS, Université Paris-Saclay, 91190 Gif-sur-Yvette, France. [2] Present address: Institut de la Vision, Univ. P. & M. Curie, INSERM, CNRS, Sorbonne Université, Paris 75012, France. These authors contributed equally: Faredin Alejevski, Alexandra Saint-Charles. Correspondence and requests for materials should be addressed to F.R. (email: rouyer@inaf.cnrs-gif.fr)

The *Drosophila* sleep–wake rhythms are controlled by a brain circadian clock that includes about 150 clock neurons[1]. Light synchronizes the clock neuronal network through cell-autonomous and non-cell-autonomous light input pathways[2,3]. Cry is a blue-light sensitive photoreceptor protein that is expressed in most clock neurons[4–8]. In the absence of Cry, flies do not phase-shift their behavioral rhythms in response to a short light pulse but still synchronize to light–dark (LD) cycles[4]. Only flies devoid of both Cry and rhodopsin-expressing photoreceptors fail to entrain to LD cycles[2,7]. Six different rhodopsins (Rhs) have been characterized in the *Drosophila* photoreceptive structures, which include the compound eye, the Hofbauer-Buchner (H-B) eyelet, and ocelli. The compound eye strongly contributes to circadian photoreception, whereas a modest contribution appears to be brought by the H-B eyelet and the ocelli[3,9–11]. A circadian function has been recently associated with the yet poorly characterized rhodopsin 7, although its exact contribution and localization in the brain and/or the eye remains controversial[12–14]. In addition to entrainment, the visual system controls other features of the clock neuron network by conveying light information to either promote or inhibit the behavioral output of specific clock neuron subsets[15–17].

The compound eye includes about 800-unit eyes (ommatidia), each of which contains eight photoreceptors. The six Rh1-expressing outer photoreceptors (R1–6) are involved in motion detection and project to the lamina neuropile of the optic lobe. The two inner photoreceptors (R7–8) are important for color detection and project to the medulla. They express four different rhodopsins and thus define two types of ommatidia: "pale" (p) ommatidia (30%) include a Rh3-expressing R7 and a Rh5-expressing R8, whereas "yellow" (y) ommatidia (70%) include a Rh4-expressing R7 and a Rh6-expressing R8[18,19]. Each extra-retinal H-B eyelet contains four Rh6-expressing photoreceptors that project to the accessory medulla, in the vicinity of key pacemaker neurons, the ventral lateral neurons (LNvs) that produce the pigment-dispersing factor (PDF) neuropeptide[9,20–24]. Each of the three ocelli contains about 80 photoreceptors that express Rh2[25]. The *Drosophila* rhodopsins cover a wide range of wavelengths from 300 nm to 600 nm[18,19], with only Rh1 and Rh6 being sensitive to red light[26].

Rhodopsin-dependent circadian entrainment involves two downstream signaling pathways, the canonical one that relies on the phospholipase C encoded by the *no receptor potential A* gene (*norpA*)[27,28] or an unknown pathway that does not contribute in very low light levels[4]. All but Rh2- and Rh5- expressing photoreceptors support synchronization in very low light[29], and at least Rh1, Rh5, and Rh6 can signal through the NorpA-independent pathway[30,31]. Photoreceptors of the compound eye are histaminergic[27,28] but the H-B eyelet expresses both histamine and acetylcholine[32,33,22]. Although the two neurotransmitters might contribute to circadian entrainment[34,35], flies devoid of Cry and histidine decarboxylase do not synchronize their rest–activity rhythms with LD cycles[10]. This suggests that besides Cry, there is no histamine-independent pathway to entrain the clock.

Two genes encoding histamine-gated chloride channels, *ora transientless (ort)* and *Histamine-gated chloride channel subunit 1 (HisCl1)*, have been identified in *Drosophila*[36–39]. The *ort*-null mutants are visually blind[40,41] and their electroretinograms have no ON and OFF transients[42]. In contrast, *HisCl1* mutants show increased OFF transients, whereas slower responses were observed in the postsynaptic laminar monopolar cells[42]. Based on transcriptional reporters, *ort* expression in the optic lobes was observed in neurons of both the lamina and medulla/lobula neuropils[41–44]. Based on reporter gene expression, *HisCl1* was localized in glial cells of the lamina[41,42]. However, recent work reported expression in photoreceptors, in particular in the R7 and

R8 inner photoreceptor subtypes[45–47]. Indeed, Ort and HisCl1 support color opponency between the two subtypes of "inner" photoreceptors, the ultraviolet (UV)-sensitive R7 and non-UV-sensitive R8, with HisCl1 and Ort mediating direct and indirect inhibition, respectively[47]. The histaminergic pathways that are involved in circadian entrainment are unknown and are the subject of the present study. Our results show that both Ort and HisCl1 define two different pathways for circadian entrainment. Whereas Ort contributes through its expression in the interneurons of the optic lobe, HisCl1 mostly contributes through its expression in the Rh6-expressing retinal photoreceptors. The work thus reveals that Rh6-expressing neurons contribute to light-mediated entrainment as both photoreceptors and interneurons.

## Results

**Ort and HisCl1 both support synchronization with LD cycles.** We asked whether Ort and HisCl1 mediate circadian entrainment. Like wild-type flies and $cry^0$ mutants, $HisCl1^{134}$ $ort^1$ double mutants resynchronized their rest–activity rhythms when the LD cycle is either advanced or delayed by several hours (Fig. 1a, Supplementary Fig. 1). In contrast, $cry^{02}$ $HisCl1^{134}$ $ort^1$ triple mutants (hereafter *CHO* mutants) did not synchronize (Fig. 1a), indicating that at least one of the two chloride channels is required in the absence of Cry. We thus tested the contribution of each of the two histamine-gated channels. Flies devoid of either Cry and HisCl1 or Cry and Ort efficiently synchronized (Fig. 1a, Supplementary Fig. 1, Supplementary Table 1, 2), showing that HisCl1 or Ort can each mediate Cry-independent entrainment of activity rhythms. Finally, we asked whether the two histamine receptors contributed to both the canonical and NorpA- independent transduction pathways[4]. As expected, $cry^0$ $norpA^{P24}$ double mutants did not synchronize with low light LD cycles (see Methods), but $cry^0$ mutants with either only HisCl1 or only Ort supported synchronization through the NorpA-dependent pathway (Supplementary Fig. 1). In the absence of NorpA and Cry, flies with one or the other receptor synchronized with high light LD cycles (Supplementary Fig. 1). Each of the two histamine receptors can thus mediate synchronization through NorpA-dependent and NorpA-independent pathways. We conclude that, in contrast to vision, circadian photoreception can thus be independently supported by Ort and HisCl1.

We tested the ability of HisCl1 and Ort to mediate circadian entrainment through inputs from either the Rh1 photoreceptors that project to the lamina or the Rh6 photoreceptors, which include retinal neurons that project to the medulla and H-B eyelet neurons that project to the accessory medulla. Since circadian entrainment to red light LD cycles (hereafter RD cycles) requires either Rh1 or Rh6[26], we used a shifted RD cycle protocol to functionally isolate these two rhodopsins. The four possible combinations of histamine receptor and rhodopsin mutants synchronized with RD cycles, although flies with only Rh1 and HisCl1 did it more slowly (Fig. 1b, Supplementary Fig. 1, Supplementary Tables 1, 2). HisCl1 and Ort could thus each respond to inputs from Rh1-expressing outer and Rh6-expressing R8 inner photoreceptors, suggesting that HisCl1 is not only expressed in the lamina.

**Ort and HisCl1 define different light input pathways.** To identify the circuits between photoreceptors and clock neurons, we asked whether targeting expression of HisCl1 or Ort in different neuronal subsets of *CHO* mutants would restore entrainment. We first used the *tim-gal4* driver, whose expression in the head includes brain clock neurons, photoreceptors, and other neurons and glial cells, in particular in the optic lobes[48].

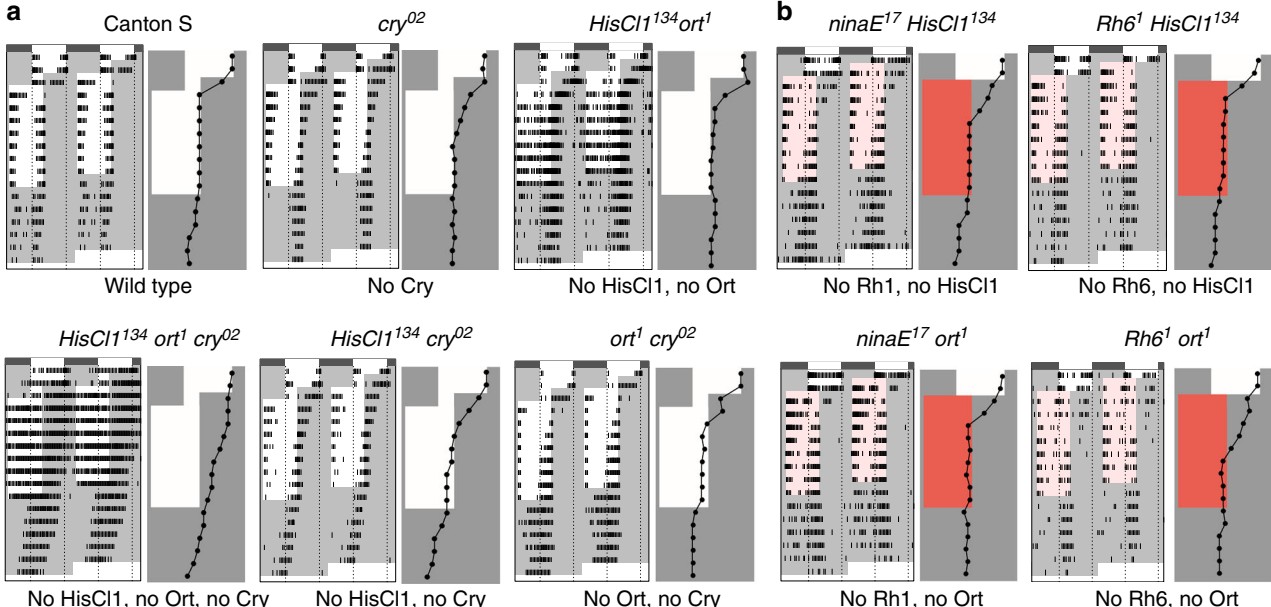

**Fig. 1** HisCl1 and Ort histamine receptors can each transmit light input from Rh1- and Rh6-expressing PRs. **a**, **b** Average double plots of locomotor activity (actograms) of flies exposed to an 8 h advance of the LD/RD cycle, with corresponding phase plots (see Methods). Flies were initially entrained with both LD and temperature cycles (TC: 25–20 °C). **a** T°C was kept constant (25 °C) from the beginning of the day 3 light phase until the end of the experiment and the day 3 light phase was shortened by 8 h. Thus, flies were exposed to a novel advanced LD regime for 8 cycles followed by constant darkness (6 DD cycles). (Top) Wild-type flies (left) as well as flies with no Cry (center) or no histamine receptors (right) synchronize with phase advanced LD cycles. (Bottom) Flies with no Cry and no histamine receptors do not synchronize (left), whereas flies with no cry and only Ort (center) or only HisCl1 (right) do synchronize. **b** T°C was kept constant (25 °C) from the beginning of the day 2 light phase until the end of the experiment. Between days 2 and 3, the dark phase was shortened by 8 h and red light was used for the advanced light phase. Thus, flies were exposed to a novel advanced RD (red light) regime for 8 cycles followed by 6 DD cycles. Each of the four genotypes with only one histamine receptor and either Rh1 or Rh6 rhodopsin synchronizes with advanced RD cycles. Bars above actograms indicate the initial LD/TC cycles (black: lights-off, 20 °C; white: lights-on, 25°). White or red areas indicate the light phase of the cycle with white or red light respectively, and gray areas the dark phase of the cycle. Dots of the phase plots indicate the peak value of evening activity in LD/RD cycles and of activity in DD. *n* is the number of flies. Some genotypes (e.g., *CHO*, in (**a**) bottom left) have a free-running period that is slightly shorter than 24 h, which results in gradual advancement of the activity peak in DD or in LD/RD (also see other figures) when not entrained

Expression of either *HisCl1* or *ort* under *tim-gal4* control restored synchronization (Fig. 2a). Expression of either histamine receptor in glial cells with the *repo* driver[49] did not rescue the *CHO* phenotype (Supplementary Fig. 2), supporting the idea that neuronal expression is required for the circadian function of both Ort and HisCl1. Further, *HisCl1-gal4*-driven glial expression[41,42] did not allow HisCl1 or Ort to restore synchronization (Supplementary Fig. 2), suggesting that *HisCl1-gal4* does not recapitulate endogenous *HisCl1* expression. Driving *ort* expression with the *ort^{c1-4}-gal4* construct that includes the four blocks of conserved sequences identified in the non-transcribed *ort* genomic region[41] allowed synchronization (Fig. 2b, Supplementary Tables 1, 2). However, *ort^{c1-4}-gal4*-driven *HisCl1* expression did not (Fig. 2b, Supplementary Tables 1, 2), indicating that HisCl1 could not functionally replace Ort in *ort*-expressing neurons. Thus, Ort and HisCl1 appear to define two different neuronal pathways for the synchronization of rest–activity rhythms.

The small and large LNvs send processes in the accessory medulla and the large LNvs also extensively arborize in the medulla, suggesting that PDF cells could receive direct inputs from inner photoreceptors and/or eyelet[9,23,34,35,50]. We thus targeted HisCl1 or Ort expression to the clock neurons of *CHO* mutants and tested entrainment. Neither HisCl1 nor Ort could restore synchronization when expressed in the PDF neurons or even with the *Clk:gal4(6/1)*[51] that is expressed in most clock neurons, including all lateral (LNvs, dorsal lateral neurons, and lateral posterior neurons) and dorsal neurons (DN1, DN2, DN3 subsets) (Fig. 2c, Supplementary Fig. 2). Our data indicate that direct histaminergic inputs from the photoreceptors to the

clock neurons are not sufficient for circadian entrainment and support an indirect connection through HisCl1- and Ort-expressing interneurons.

The failure of HisCl1 to allow synchronization when expressed in Ort-expressing neurons prompted us to look for other types of neurons that potentially express HisCl1. Surprisingly, targeting HisCl1 to all photoreceptors with *GMR-gal4* rescued the entrainment of *CHO* mutants (Fig. 2d, Supplementary Table 1, 2). Furthermore, rescue was no longer observed in *tim>HisCl1, CHO* flies when *HisCl1* expression in photoreceptors was blocked in the photoreceptors by concomitantly expressing the Gal4 inhibitor Gal80 under *GMR* control (Fig. 2d, Supplementary Table 1, 2). This raised the intriguing possibility that HisCl1 circadian function occurs in photoreceptor cells.

**HisCl1, but not Ort, is expressed in photoreceptors**. Transcriptome analysis indicated that *HisCl1* but not *ort* was expressed in photoreceptors[45,46]. In agreement with previous studies[41–43], we could not detect any photoreceptor-localized expression of the different *ort-gal4* drivers (not shown). The published anti-HisCl1 antibody[43] only produced non-specific labeling in our hands and a *HisCl1-gal4* construct showed expression only in the optic lobe, essentially in lamina cells as described previously[41]. However, by using two copies of a second *HisCl1-gal4* construct[42], we could faintly detect some inner retinal photoreceptors and the eyelet, in addition to the lamina cells (Fig. 3a–d). The expression of *HisCl1* in photoreceptors was also faintly detected by a *HisCl1::egfp* protein fusion genomic

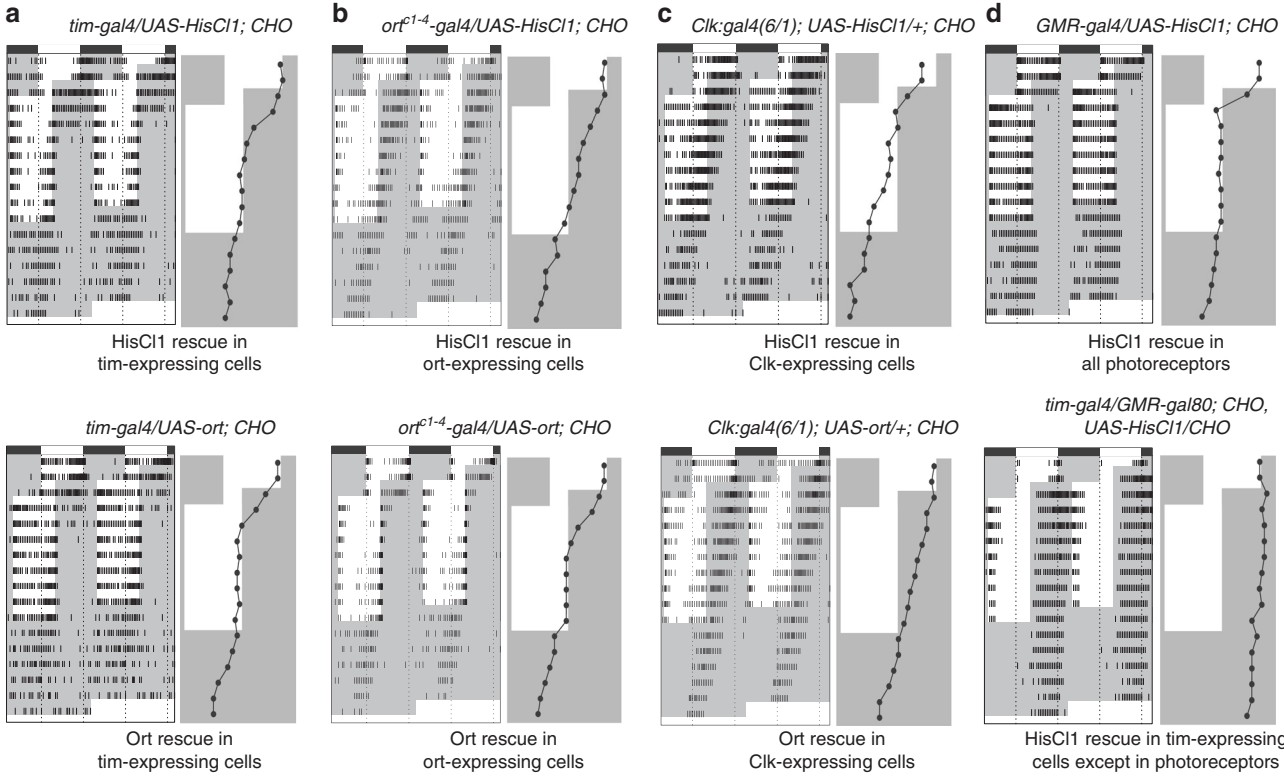

**Fig. 2** Ort and HisCl1 define different neuronal pathways for clock synchronization. Actograms and phase plots of flies with expression of the two histamine receptors in the triple mutant (*HisCl1^{134} ort^1 cry^{02}–CHO*) genetic background using different *gal4* drivers. The experimental design is as described in Fig. 1 legend. **a** Expression of either *HisCl1* (top) or *ort* (bottom) in *tim*-expressing cells rescues entrainment. **b** *HisCl1* cannot rescue entrainment when expressed in *ort*-expressing neurons (top), whereas *ort* can (bottom). **c** Neither *HisCl1* (top) nor *ort* (bottom) rescues entrainment when expressed in most of the clock neurons. **d** Entrainment is rescued by *HisCl1* expression in photoreceptors (top) and is not rescued by *HisCl1* expression in *tim*-expressing cells when expression in the photoreceptors is blocked by the Gal4 inhibitor Gal80 (*GMR-gal80*) (bottom)

construct (see Methods), which labeled the cell bodies of both a fraction of R7 and R8 inner retinal photoreceptors and eyelet photoreceptors (Fig. 3e–g). *HisCl1::egfp* expression was recently reported in R7 and R8 inner photoreceptors[47]. To confirm *HisCl1* reporter gene expression data, we addressed the retinal expression of the *ort* and/or *HisCl1* genes with a new sensitive in situ hybridization technique (see Methods). Indeed, *HisCl1* messenger RNA (mRNA) was present at low levels in photoreceptors of wild-type flies, including Rh6-expressing R8 cells, whereas no transcripts were detected in *Hiscl1^{134}* mutants (Fig. 3h–k) or with a negative control probe (not shown). In contrast, *ort* expression was observed in interneurons but not in photoreceptors (Fig. 3l, m).

**HisCl1 circadian role is in Rh6-expressing photoreceptors.** The expression data suggested that HisCl1 circadian function could map to inner photoreceptors of the retina and/or eyelet. Among inner photoreceptors, R7 cells express Rh3 or Rh4, whereas R8 cells express Rh5 or Rh6[18,19,52]. We thus restored HisCl1 in *CHO* mutants with different types of *Rh-gal4* constructs. A fast synchronization with the shifted LD cycle was observed with *Rh6-gal4*, which was lost in the presence of *GMR-gal80* (Fig. 4a, Supplementary Tables 1, 2). In contrast, none of the other *Rh-gal4* rescued entrainment (Supplementary Fig. 3, Supplementary Tables 1, 2), showing that HisCl1 acts specifically in Rh6-expressing photoreceptors. Of course, we cannot exclude a possibility that synchronization rescue could occur over longer exposure to the LD/RD cycle. We also observed that this new role of Rh6 cells could mediate circadian entrainment through both

NorpA-dependent photoreception in low light and NorpA-independent photoreception in high light (Supplementary Fig. 3, Supplementary Tables 1, 2). Surprisingly, expressing Ort in Rh6 cells also rescued entrainment (Supplementary Fig. 3, Supplementary Tables 1, 2), indicating that Ort can replace HisCl1 function in the photoreceptors, whereas HisCl1 could not replace Ort in the *ort*-expressing interneurons of the optic lobe (see Fig. 2a).

What is the specific contribution of HisCl1 function in Rh6 photoreceptors to circadian entrainment? To address this question, we first downregulated *HisCl1* in Rh6 cells, with other putative sites of *HisCl1* expression being not affected. *HisCl1* RNA interference (RNAi) was expressed in a *cry^0 ort^1* double mutant background and the flies were tested in RD cycles to restrict light inputs to Rh1 and Rh6 cells. *HisCl1* RNAi flies did synchronize with the new light regime but substantially slower than the control flies, supporting a contribution of *HisCl1* expression in Rh6 photoreceptors to entrainment (Fig. 4b, Supplementary Fig. 4 and Supplementary Tables 1, 2). The fact that *HisCl1* RNAi flies still synchronize, although slowly, could result from an incomplete loss of HisCl1 function in Rh6 photoreceptors and/or from the existence of a photoreceptor-independent role of HisCl1. A strong contribution of HisCl1 in photoreceptors was confirmed by testing flies that were mutant for *HisCl1* in whole eye clones (see Methods). The synchronization with advanced LD cycles was strongly slowed down (Fig. 4c, Supplementary Fig. 4, Supplementary Table 1), although our statistical test failed to capture a significant difference with controls in the last 5 days of the shifted LD cycle (Supplementary Table 2). However, flies with *HisCl1^{134}* mutant eyes failed to

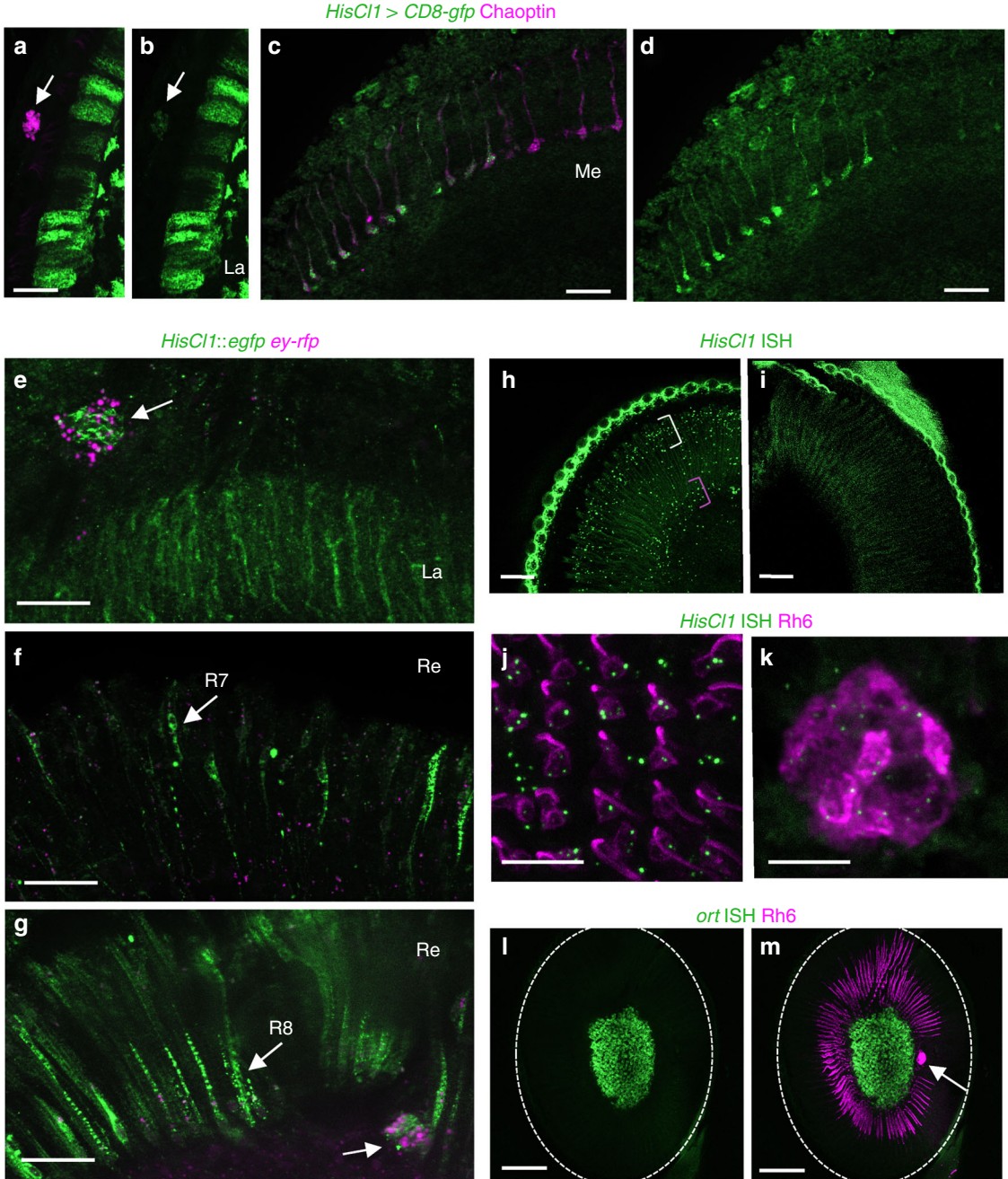

**Fig. 3** HisCl1 but not Ort is expressed in photoreceptors. **a–d** Expression of *HisCl1-gal4* UAS-*mCD8-gfp*, detected with anti-GFP antibody (green); photoreceptors are marked with anti-Chaoptin antibody (red). *HisCl1* expression is detected in the lamina epithelial cells and very faintly in the eyelet (**a**, **b**). Weak labeling is observed in some R7 and R8 photoreceptors axons in the medulla (**c**, **d**). **e–g** Expression of GFP-tagged HisCl1 protein expressed from the *HisCl1* promoter detected with anti-GFP antibody (green); photoreceptors are marked with ey-*rfp* from the same fosmid construct (red). HisCl1::GFP is detected in the lamina (**e**), H-B eyelet (**e**, **g**), and both in the R7 (**f**) and R8 (**g**) photoreceptors, identified by their respective position in the retina (arrows), upper layer for R7 and lower layer for R8. **h–m** *HisCl1* (**h-k**) and *ort* (**l**, **m**) mRNA expression (blue) visualized by RNAscope® in situ hybridization (ISH) (see Methods). Anti-Rh6 antibody (red) labels yR8 and eyelet photoreceptors in (**j**, **k**, **m**). *HisCl1* mRNA in the retina appears as puncta primarily in two layers: proximally, near the R8 photoreceptor nuclei (**h** pink bracket) and distally, near the R1-7 photoreceptor and other retinal cell nuclei (**h** yellow bracket). No puncta were detected in the *HisCl1^134* mutant (**i**). In the R8 photoreceptor layer *HisCl1* mRNA is expressed in the Rh6-positive (red) yR8 retinal photoreceptors, as well as in the surrounding cells that likely include Rh5-expressing pR8 photoreceptors (**j**). *HisCl1* mRNA is also present in the Rh6-expressing H-B eyelet (**k**). *ort* mRNA is expressed in the lamina but not in the retina (**l**, **m**). In particular, *ort* is undetectable in the Rh6-expressing cells (red in **m**) both in the retina (outlined in **l**, **m**) and in the eyelet. In (**h**, **i**), the retinas are surrounded by the autofluorescent cuticle. Images in (**h–m**) are maximal projections of short confocal stacks (thickness: 10 (**h**, **i**), 12 (**j**, **k**), 13.5 (**l-m**) μm). La lamina, Me medulla, Re retina. White arrows point to the H-B eyelet. Scale bars represent 20 (**a–i**), 10 (**j**, **k**), and 50 (**l**, **m**) μm

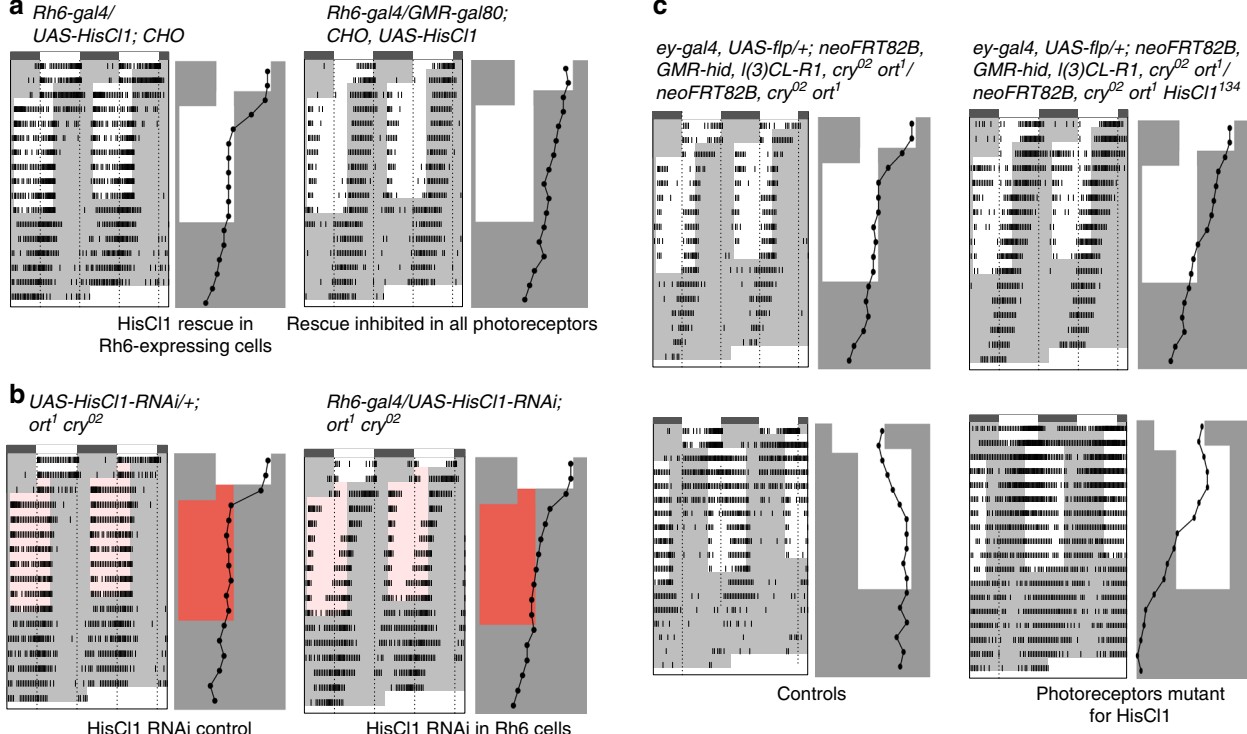

**Fig. 4** HisCl1 acts in Rh6-expressing photoreceptors to support circadian entrainment. Actograms and phase plots. The experimental design is as described in Fig. 1 legend for advanced LD/RD cycle (**a**, **b**, **c**, top). For delayed LD cycles (**c**, bottom), the dark phase was lengthened by 8 h between days 2 and 3. **a** Expression of *HisCl1* specifically in Rh6 photoreceptors with a *Rh6-gal4* driver rescues entrainment in *CHO* triple mutants (left). This rescue is blocked by simultaneous expression of Gal80 in all photoreceptors (right). **b** In the absence of Ort and Cry, RNAi knock-down of *HisCl1* in Rh6 photoreceptors slows down the synchronization with advanced RD cycles (right), in comparison to controls (left). **c** In contrast to control flies with *HisCl1+* photoreceptors (left), flies with *HisCl1^{134}* mutant photoreceptors (right) very poorly synchronize with an advanced LD cycle (top) and fail to synchronize with a delayed LD cycle (bottom)

synchronize with a delayed LD cycle, supporting a critical eye contribution for HisCl1 function. Altogether, the results indicate that photoreceptors are the major site of HisCl1 expression for synchronization with LD cycles.

**Rh6-expressing cells receive multiple photoreceptor inputs.** Finally, we investigated the histaminergic input pathways relevant for HisCl1 in Rh6-expressing cells. Flies with HisCl1 only in Rh6 cells synchronized with RD cycles (Fig. 5a, Supplementary Tables 1, 2), indicating that Rh6 cells can rely on histaminergic input from photoreceptors expressing Rh1 and/or Rh6 to support entrainment. They synchronized more slowly in the absence of Rh1 (Fig. 5a, Supplementary Fig. 4), suggesting contribution from Rh1 cells. Nevertheless, their ultimate synchronization indicates that inputs from at least some Rh6-expressing photoreceptors contribute to HisCl1 function in Rh6 cells. No synchronization occurred when HisCl1 was restored in Rh6 cells of *CHO* mutants in which the compound eye, but not the eyelet, was lost as a consequence of a null mutation in the *sine oculis* gene (Fig. 5b, Supplementary Tables 1, 2). Thus, if the Rh6-expressing eyelet is involved in this HisCl1 pathway, it would require inputs from retinal photoreceptors. In addition, genetic ablation of the adult Rh5-expressing cells as well as of the eyelet (see ref. [24]) in a *CHO* background with HisCl1 rescue in Rh6 cells did not abolish entrainment in RD cycles (Fig. 5b, Supplementary Table 1, 2). This indicated that the eyelet is not required for HisCl1 function in Rh6 photoreceptors through Rh1 and/or Rh6 inputs. We then asked whether HisCl1 function in Rh6 cells would require their ability to act as photoreceptors. Since HisCl1 in Rh6-expressing cells could rescue entrainment in the absence of Rh6 (Fig. 5c,

Supplementary Tables 1, 2), we can conclude that the interneuron role of Rh6 cells does not require their photoreceptive function. Importantly, efficient synchronization still occurred in the collective absence of Rh1, Rh5, and Rh6 (Fig. 5c, Supplementary Tables 1, 2). Although a contribution of the dorsally located Rh2-expressing ocelli cannot be totally excluded, this result strongly suggests that R7 photoreceptors, which express UV-sensitive Rh3 or Rh4, also provide inputs to HisCl1 in Rh6-expressing photoreceptors.

## Discussion

This work reveals that the Cryptochrome-independent entrainment of rest–activity rhythms relies on distinct pathways that are contributed by the two histamine receptors Ort and HisCl1. Whereas Ort mediates circadian entrainment through the optic lobe interneurons that are involved in visual functions, HisCl1 defines a new photoreceptive pathway through Rh6-expressing photoreceptors. Although both receptors mediate synchronization with a shifted LD cycle, it seems likely that the two pathways will show differences in specific light conditions. We could not rescue Ort function with *HisCl1* expression in the *ort*-expressing cells, whereas the Ort could replace HisCl1 in Rh6 photoreceptors. It is possible that HisCl1 has a lower affinity for histamine with Rh6 cells receiving more neurotransmitter than optic lobe interneurons. Alternatively, interneurons could sufficiently differ from photoreceptors for their physiology or specific receptor-interacting protein content, preventing HisCl1 from working efficiently. HisCl1 downregulation in Rh6 cells slows down synchronization and flies with *HisCl1^{134}* mutant eyes synchronize very poorly with advanced LD cycles, and fail to

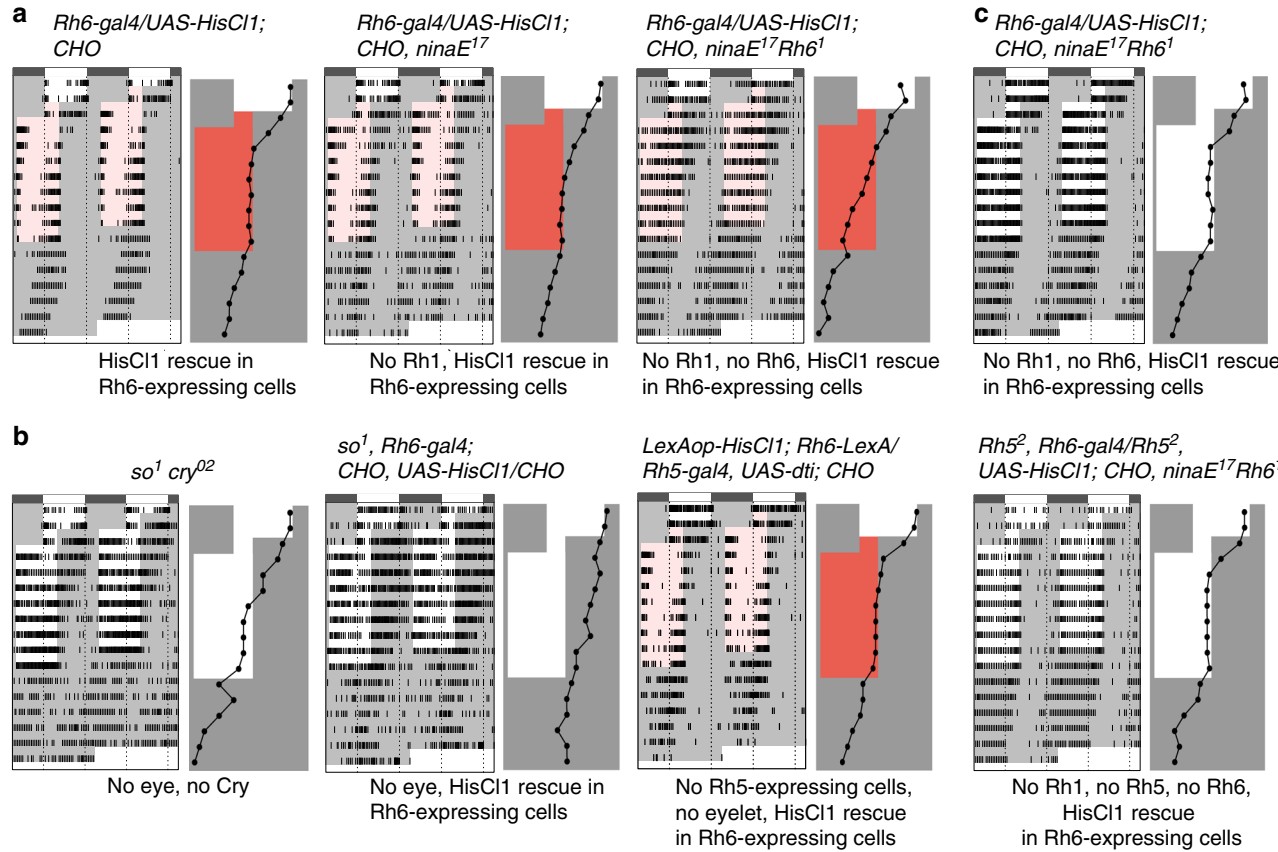

**Fig. 5** HisCl1 in Rh6-expressing photoreceptors receive inputs both from outer and inner photoreceptors. **a–c** Actograms and phase plots. The experimental design is as described in Fig. 1 legend. **a** The RD phase advance rescue of *CHO* flies by *HisCl1* expression in Rh6 photoreceptors (left) is slowed down by the absence of Rh1 (*ninaE^17* mutation, center) and is abolished by the absence of both Rh1 and Rh6 (*ninaE^17 Rh6^1* genetic background, right). **b** Flies lacking Cry, compound eyes, and ocelli but retaining the H-B eyelet synchronize with advanced LD cycles (*so^1; cry^02* double mutants, left), but do not if *HisCl1* expression is restricted to Rh6 photoreceptors (center). CHO mutants with HisCl1 rescue in Rh6 cells and genetic ablation of the eyelet (in addition to Rh5-expressing photoreceptors) synchronize with advanced RD cycles (right). **c** The LD phase advance rescue of *CHO* flies by *HisCl1* expression in Rh6 photoreceptors is retained in the absence of either Rh1 and Rh6 (top) or Rh1, Rh5, and Rh6 (bottom)

synchronize with delays. We cannot exclude that non-photoreceptor cells contribute to HisCl1-dependent entrainment but other pathways appear to have a modest contribution if any.

HisCl1 is expressed in the H-B eyelet, which could thus contribute to this synchronization pathway. However, the cell-killing experiments indicate that H-B eyelet is not required for HisCl1-mediated synchronization through Rh6 cells. In the recently described color opponency mechanism, retinal R7 cells inhibit R8 and vice versa through HisCl1 expression in the photoreceptors[47]. We suppose that HisCl1-dependent clock synchronization is also mediated by the hyperpolarization of Rh6-expressing cells. How this hyperpolarization interacts with the light-induced depolarization in Rh6 photoreceptors to result in a synchronization message to the clock neurons remains to be understood. Since only Rh6-expressing R8 and not the other inner photoreceptors contribute to this circadian photoreception pathway, Rh6 cells might have specific connections with downstream interneurons. Such specificity has been described for color vision where each of the four inner photoreceptor subtypes connects to a different type of TmY interneuron in the Medulla[53]. We show that HisCl1 expression in Rh6 cells supports synchronization with red light, in the absence of Rh1, indicating that an intra-Rh6-photoreceptor circuit is sufficient. This indicates that Rh6-expressing R8 photoreceptors play a dual photoreceptor/interneuron role in this pathway (Fig. 6). Whether the same individual cells have the two roles is unknown, although the HisCl1-dependent color

opponency mechanism suggests that it could be the case. It is also unclear whether all Rh6-expressing R8 photoreceptors or only a fraction of them contribute to circadian synchronization. Our results imply that, in addition to histaminergic neurotransmission, Rh6-expressing photoreceptors can talk to downstream interneurons through histamine-independent neurotransmission (Fig. 6). A recent transcriptomics study indeed revealed the expression of cholinergic markers in R7 and R8 cells, supporting cholinergic transmission in the inner photoreceptors, in addition to histaminergic transmission[46].

Our data indicate that histaminergic inputs from both outer and inner photoreceptors converge to Rh6 cells to contribute to circadian entrainment. It is possible that some of these inputs rely on Rh7, which seems to be expressed in Rh6-expressing photoreceptors, according to transcriptional reporter data[14]. Putative connections between photoreceptors have been described in *Drosophila*[47,54,55] and other insects[56,57]. How R1–6 photoreceptors might be connected to Rh6-expressing R8 cells remains uneasy to understand, but a few putative contacts between presynaptic outer cells and postsynaptic inner cells have been observed in *Musca*[56]. The intra-retinal functional connectivity that we report in *Drosophila* is reminiscent to the circuit logic of circadian entrainment in the mammalian retina, where intrinsically photoreceptive retinal ganglion cells express the melanopsin photopigment in addition to receiving inputs from rods and cones[58–60]. Interestingly, melanopsin appears to share light-

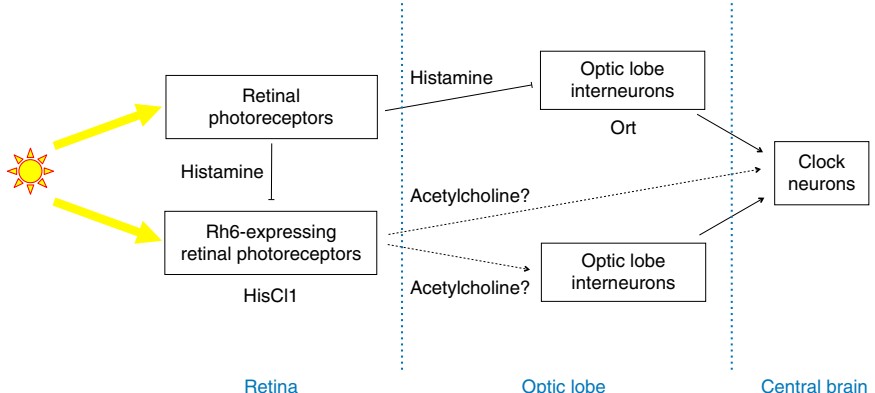

**Fig. 6** Model for the retinal input pathways to the brain clock. All retinal photoreceptors receive light inputs and release histamine. Ort-expressing interneurons of the optic lobe receive inhibitory histamine signals from most or all rhodopsin-expressing photoreceptors and transmit light information to the clock neurons through direct or indirect pathways. At least some of the Rh6-expressing photoreceptors express HisCl1 and receive histamine signals from other photoreceptors, in addition to receiving direct light inputs. These Rh6-expressing photoreceptors use a histamine-independent neurotransmission (possibly cholinergic) to transmit light information to the clock neurons. This connection could be either direct or via interneurons of the optic lobe

sensing properties with the rhabdomeric photoreceptors of invertebrates[61,62]. It has been shown that the mammalian circadian clock can synchronize with day–night cycles by tracking light color changes in addition to light intensity changes[63,64]. It will be interesting to investigate the possible contribution of the dual function of Rh6-expressing photoreceptors to integrating different color cues into the retinal information that is sent to the clock.

## Methods

**Fly stocks**. The *HisCl1::egfp, ey-rfp* fosmid (see ref. [65]) contains a large genomic DNA fragment encompassing the *HisCl1* gene (VDRC 318735). Two different *HisCl1-gal4* constructs were used, which slightly differ in the length of the 5' sequences driving *gal4* expression[41,42]. A list of all stocks is provided in the accompanying supplementary information.

**Generation of transgenic lines**. To produce *UAS-HisCl1* flies, we prepared mRNA from *Canton S* flies, amplified the open reading frame (ORF) of the *HisCl1* gene with reverse transcription (RT)-PCR. The following primers were used for RT-PCR: forward, 5'-CACCATGCAAAGCCCAACTAGCAAATT-3' and reverse, 5'-TCATAGGAACGTTGTCCAATAG-3'. The resulting complementary DNA (cDNA) was cloned into an entry TOPO vector (pENTR/D-TOPO, Invitrogen). The *UAS-HisCl1* transgene was generated by recombining the *HisCl1* ORF into a destination pTHW vector (Drosophila Gateway System) in frame with a N-terminal 3xHA epitope tag. Then this construct was introduced into fly by standard insertion of a P-element (Fly Facility, France). One insertion on the second chromosome and one on the third chromosome were used in this study. Both rescue entrainment of *CHO* mutants when driven by *tim-gal4*. *LexAop-HisCl1* flies were generated by amplifying the HA-HisCl1 coding region from the *w; UAS-HA-HisCl1* line with forward primer 5'-*tat*CTCGAGATGGATCTCCACCGCGGT-3' and reverse primer 5'-aa*TCTAGA*TCATAGGAACGTTGTCCAATAGACA-3'. The plasmid pJFRC19-13XLexAop2-IVS-myr::gfp (Addgene #26224) was digested with *Xho*I and *Xba*I to excise the Myr::gfp sequence and replace it with HA-HisCl1 amplicon previously digested with *Xho*I and *Xba*I. After sequencing the coding region, the construct was introduced inside the X chromosome attP8 docking site doing a phiC31 integrase-mediated transformation (BestGene).

**Generation of *HisCl1* mutant eye clones**. Flies with *HisCl1[134]* eye clones were generated by using *eyeless*-driven flippase recognition target (FRT)-mediated mitotic recombination[66]. The stocks required for generation of the somatic clones of the eye were obtained from Bloomington Drosophila Stock Center (BDSC). BDSC#5253 containing *eyeless-gal4, UAS-flp* on the second chromosome, and *neoFRT[82B]* GMR-*hid, l(3)CL-R[1]* on the third chromosome was crossed with *cry[02] ort[1]* flies to put the different transgenes in a double mutant background by genetic recombination. BDSC#2035 containing *neoFRT[82B]* on the third chromosome was used to put this transgene in a *cry[02], ort[1]* background or in a *cry[02], ort[1], Hiscl1[134]* background. The three lines were PCR-controlled for the presence of *neomycin*R, *cry[02], ort[1]*, and *Hiscl1[134]* when applicable. The *ey-gal4*-induced FRT recombination produced flies with a close to wild-type eye morphology in the presence of GMR-*hid, l(3)CL-R[1]*, indicating that vast majority of photoreceptors were homozygous

for the *Hiscl1[134]* mutation. The final genotypes were *w; ey-gal4, UAS-flp/+; neoFRT82B, GMR-hid, l(3)CL-R[1], cry[02] ort[1]/neoFRT82B, cry[02] ort[1]* (control) and *w; ey-gal4, UAS-flp/+; neoFRT[82B], GMR-hid, l(3)CL-R[1], cry[02] ort[1]/neoFRT[82B], cry[02] ort[1] Hiscl1[134]* (mutant).

**Behavioral analysis**. Experiments were conducted with 3–5-day-old males placed in *Drosophila* activity monitors (TriKinetics) as described in ref. [29]. In all experiments, flies were first entrained for 2 days under LD + T°C cycles 12 h light (25 °C)–12 h dark (20 °C). Light was provided by light-emitting diodes (LEDs; 400 nm−680 nm at ~2000 lux). For standard (high light) LD (light–dark) phase advances, T°C was kept constant (25 °C) from the beginning of the day 3 light phase until the end of the experiment and the day 3 light phase was shortened by 8 h, followed by LD 8 cycles (LED source, 400 nm−680 nm at ~2000 lux), then constant darkness (6 DD cycles). For low light LD advances, the 8 advanced LD cycles were done with dim light (LED source, 325 nm−625 nm at ~5 lux). For RD (red light–dark) advances (LED source, 590 nm−680 nm at ~250 lux), short-night or short-day protocols were used and gave the same results. For short night (Fig. 1 and Supplementary Fig. 1), T°C was kept constant (25 °C) from the beginning of the day 2 light phase until the end of the experiment. Between days 2 and 3, the dark phase was shortened by 8 h, followed by RD 8 cycles, then constant darkness (6 DD cycles). For short days (Figs. 3 and 4), T°C was kept constant (25 °C) from the beginning of the day 3 light phase until the end of the experiment. The day 3 light phase was shortened by 8 h and red light was used instead of white light, followed by RD 8 cycles, then constant darkness (6 DD cycles). For standard (high light) LD phase delays, T°C was kept constant (25 °C) from the beginning of the day 2 light phase until the end of the experiment. Between days 2 and 3, the dark phase was lengthened by 8 h followed by LD 8 cycles (400 nm−680 nm LEDs at ~2000 lux), then constant darkness (6 DD cycles). Data analysis was done with the FaasX 1.21 software, which is derived from the Brandeis Rhythm Package[67,68]. FaasX runs on Apple Macintosh OSX and is freely available (http://neuro-psi.cnrs.fr/spip.php?article298&lang=en). Bin size was 30 min. The data are shown as average double-plotted actograms from *n* flies. All behavioral experiments were done at least three times (except when indicated) with similar results. The statistical analysis of the phase data was done as follows. For each day, the ZT that corresponds to the peak value of evening activity (second half of the day) in LD/RD cycles was determined from the averaged activity data (*n* flies). These values were used to generate phase plots that are shown next to the corresponding actograms to better visualize the daily phase during all experiments. Entrainment was then estimated by comparing the peak ZT values of the last 5 days of the shifted LD/RD regime to those of Canton S controls. A genotype was considered entrained when this series of five values was not significantly different from the control one in a one-way analysis of variance (ANOVA) Dunnett's multiple comparisons test (GraphPad Prism 7 software). Results of the ANOVA are shown in Supplementary Table 2. Of course, we cannot exclude that genotypes that are not synchronized during the 8-day interval would do it over a longer interval, or in another experimental paradigm. The period of free-running rhythms in DD was estimated by $x^2$ periodogram analysis from the last 5 days of data with the following criteria for rhythmic flies (filter ON): power ≥ 20, width ≥ 2 h. Power and width correspond to the height and width of the periodogram peak, respectively, and give the significance of the calculated period.

**Immunolabelings**. Brains with retinas were dissected, fixed, and labeled as previously described[69]. Briefly, we dissected brains with the retinas attached to the

optic lobes and incubation with the primary antibodies anti-GFP (chicken, Invitrogen) and anti-Chaoptin (mouse, clone 24B10, gift from A. Hofbauer), was done at least 48 h at 4 °C, followed by PBT (phosphate buffer saline (PBS) + 0.1% Tween-20) washes and incubation with secondary antibodies, anti-chicken Alexa-647 conjugated (1:1000) and Alexa-488 conjugated (1:1000) and anti-mouse FP547-conjugated (1:1000) for at least 24 h at 4 °C. The samples were imaged with a Zeiss AxioImager microscope with an ApoTome structured illumination module.

**In situ hybridization**. In situ hybridization was performed with the RNAscope® Multiplex Fluorescent Reagent Kit v2 (ACD Bio)[70,71] using a protocol optimized for whole mount *Drosophila* adult retina staining. Unless specified otherwise, the solutions supplied with the kit were used. The retinas with or without the attached laminas were dissected in PBS as in ref. [72], fixed in paraformaldehyde 3.7% for 1 h at room temperature (RT°C) and washed with PBT. The tissue was treated with 3% hydrogen peroxide (3% $H_2O_2$) for 20 min at RT°C. The target retrieval treatment was done for 2 min at 100 °C in 1× Target Retrieval solution. Retinas were then treated with undiluted Protease IV for 30 min at RT. Each of the preceding steps was followed by two 10 min washes in PBT+1% bovine serum albumin. Retinas were transferred to probe diluent solution. The probes were warmed to 40 °C and added to the samples: *HisCl1* probe (Cat. No. 300031-C3 created by ACD Bio for the present study) was diluted (1:50 in probe diluent), whereas the *ort* probe (Cat. No. 435481, ACD Bio) was undiluted. After an overnight incubation at 40 °C, the samples were washed twice 10 min with 1× Wash Buffer and incubated with 2–3 drops of RNAscope® Multiplex FL v2 at 40 °C for 30 min and then washed as above. These steps were repeated with RNAscope® Multiplex FL v2 Amp 2 and then RNAscope® Multiplex FL v2 Amp 3 incubations. The samples for *ort* and negative control stains were incubated with RNAscope® Multiplex FL v2 HRP-C1, and for *HisCl1* stains with RNAscope® Multiplex FL v2 HRP-C3 for 15 min at 40 °C, followed by washes as before. Finally, the samples were incubated for 30 min at 40 °C with Opal 520 (Perkin Elmer, FP1487001KT, 1:2000 dilution) for *ort* and negative control probes and Opal 650 (Perkin Elmer, FP1496001KT, 1:2000 dilution) for *HisCl1* probe. The samples were washed as above and incubated overnight at 4 °C with anti-Rh6 antibody (rabbit, 1:1000)[73], washed in PBT, and incubated with a secondary Alexa555-conjugated donkey anti-rabbit antibody (Invitrogen, Cat. No. 1891766, 1:500) for 3 h at RT°C. Retinas were mounted on glass slides with SlowFade™ Gold antifade reagent (Invitrogen) as in ref. [72]. The samples were imaged with SP8 Leica Confocal Microscope.

## Data availability
The datasets generated and analyzed during the current study are available from the corresponding author on reasonable request.

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

## Acknowledgements

We thank the Bloomington Drosophila Stock Center and the Vienna Drosophila Research Center (VDRC) for fly stocks, M. Boudinot for the FaasX software, as well as A. Chatterjee and J. Carcaud for discussions and insightful comments on the manuscript. D.V. acknowledges funding from European Union 7th Framework Program (Marie Curie Career Integration Grant). F.R. acknowledges funding from Agence Nationale de la Recherche (ClockEye and TEFOR grants), Fondation pour la Recherche Médicale (Equipe FRM grant), and European Union 6th (EUCLOCK Integrated Project) and 7th (INsecTIME Initial Training Network) Framework Programs. D.V. and F.R. are supported by Institut National de la Santé et de la Recherche Médicale (INSERM).

## Author contributions

F.A., A.S.-C., C.M.-V. and F.R. designed the project. F.A., A.S.-C., C.M.-V., B.M., S.G. and D.V. designed and performed the experiments. F.R. wrote the manuscript with input from F.A., A.S.-C., C.M-V., and D.V.

## Additional information

**Competing interests:** The authors declare no competing interests.

