## [Peer Review File · Nature Communications]

Reviewers' comments:

Reviewer #1 (Remarks to the Author):

This is a terrific contribution. The scale and scope of the genetic experiments is remarkable. These make it possible to better understand CRY-independent entrainment pathways. They distinguish roles for two different histaminergic pathways in the fly eye/brain. They lead to the fascinating model in which (akin to the design for mammalian photoentrainment) a key photoreceptor class (Rh6-expressing) provides entrainment information to central pacemakers by incorporating synaptic inputs from other photoreceptors. The work is novel, well presented and of interest to a broad audience. I have one general concern about the analysis of behavioral synchronization as described below.

My main (and only) difficulty concerns the definitions by which flies were scored for light synchronization. No strict definition or quantitative landmark is applied; instead the authors offer a subjective determination based on the phase of the evening activity. In so doing the authors are in fact well within the traditions of field. However, I recommend in this case a more objective measure could be valuable.

Early on in this manuscript, rhythmic behavior is scored either “synchronized” or “not”, as in Figure 1. In Fig 1B all four genotypes are said to show “efficient synchronization” to the light conditions. However, I find that the Rh6[1], ort[1] combination entrains poorly compared to the other three: it appears to need 4 (or maybe even 5) cycles to finally achieve proper register compared to others.

Later, the authors also discriminate on the basis of how quickly flies display synchronization: “Control flies phase-shifted within two days but HisCl1 RNAi flies took about five days to synchronize with the new light regime (Fig. 3N). We conclude that in such conditions, HisCl1 is required for fast synchronization with LD cycles.”

Also in Fig 4A, the introduction of an Rh1 mutation is said to slow synchronization. But the basis by which the speed is judged in this case (and not in Figure 1) is unclear.

Finally, on p. 5: “In contrast, none of the other Rh-gal4 rescued entrainment (ED Fig. 3), showing that HisCl1 acts specifically in Rh6-expressing photoreceptors.” To my eye, Rh4 provided entrainment

after 5 days - slow but clear. Was a minimum number of synchronized cycles required? In the absence of a quantitative definition for synchronization, the basis for many conclusions are unclear.

Minor points:

1. "In contrast to vision, circadian photoreception can thus be independently supported by Ort and HisCl1".

Agreed, but their equivalence could be a function of a laboratory setting: they may not be completely redundant in nature

2. Finally, we addressed the retinal expression of the ort and/or HisCl1 genes with a new sensitive in situ hybridization technique.

Suggest adding a citation to this statement .

3. I found no Methods section or even a reference to a section on-line. Is Methods description limited to the Figure Legends? I guess this is a question for the Editors.

Reviewer #2 (Remarks to the Author):

This paper, along with a recently published Cell paper on color opponency (Ref 17), shows that there is communication between photoreceptors in the Drosophila retina. In this case, it is to to entrain the circadian clock. Thus, this shows that the fly retina, like its mammalian counterpart, contributes to visual processing rather than simply detecting light.

The nature of clock entrainment by light in flies has been a vexing problems for years since it appears that there is extensive redundancy, with Cryptochrome in clock neurons, photoreceptors in the retina and the eyelet, a remnant of the larval eye that, with its four photoreceptors, also contributes to light entrainment. This means that it takes multiple mutants to dissect the process. To make things worse, a recent report showed that an atypical rhodopsin, Rh7, is also expressed in clock neurons and participates to clock entrainment through a non-classical phototransduction

cascade. Strangely though, most recent papers on the clock appear to ignore Rh7, and the current paper does as well.

Photoreceptors are activated by light but are histaminergic and therefore, inhibit their targets in response to light. Mammalian photoreceptors do the exact opposite. Since most of the histaminergic neurons are the photoreceptors, it is often assumed that any cell that expresses the Histamine receptor Ort is a target of photoreceptors. However, there is another understudied Histamine receptor, HCl1 that was totally ignored until the recent report that shows that it is involved in synapses between R7 and R8 for color opponency. The current paper shows that photoreceptors also talk to one another, but in this case to mediate clock entrainment. The authors show that the Rh6-expressing green photoreceptors receive input from outer photoreceptors as well as from UV-sensitive R7 and through a mysterious way, instruct the clock: this means that R8 serves as an interneuron by collecting information from other photoreceptors. The synapses between photoreceptors are not using Ort but instead, the special histamine receptor HCl1 (as for color opponency) to entrain the clock: therefore, this is the second example of local processing in the fly retina, which is highly unexpected and exciting.

The paper describes a large number of manipulations with often very complex genotypes and these indicate that the model is likely to be correct.

However, there is an obvious and critical experiment that is missing that makes me believe that things might not be so simple: deleting or silencing (or activating since Histamine is inhibitory) the Rh6 expressing R8 cells should suppress entrainment and demonstrate its role as an interneuron. The authors must have performed this very simple experiment (which they have done for Rh5-expressing R8!).

They also argue that outer photoreceptors (as well as R7) signal to Rh6-expressing R8. However, synapses between outer and R8 photoreceptors have not been reported in the connectome from *Janelia*. The role of R7 is also not tested although it would be a trivial experiment to do so (using sevenless mutants).

Therefore, showing (simple) processing in the fly retina is an important finding that would reinforce the data published for color vision and would have significant evolutionary implications.

Once the missing experiments are performed (delete Rh6-expressing R8, use sev mutants, eventually show that outer photoreceptors synapse to R8), the paper should be published as a very significant piece of work.

Reviewer #3 (Remarks to the Author):

Review for Alejevski et al.

This is a very interesting paper from the Rouyer group about the role of the 2 histamine-gated Chloride channels encoded in the *Drosophila* genome in synchronizing the circadian clock to light-dark cycles. It is well known that light-resetting in *Drosophila* involves the blue-light photoreceptor Cryptochrome (Cry) expressed in about 50% of the clock neurons as well as visual receptors in the compound eye. Removing both Cry and histamine—the principal neurotransmitter of the visual system, results in flies that can no longer be synchronized to light. But how the visual system signals to the clock neurons is so far not know. This study supplies a first attempt by resolving how and where the 2 histamine receptors function in light resetting. The authors show that both receptors contribute to light resetting and that the relevant expression of HisCl1 occurs in the Rh6 expressing R8 photoreceptor cells, whereas that of Ort is in the optic lobe interneurons. Interestingly, Ort can replace HisCl1 function in the R8 cells but not vice versa. The authors then go on to show that—very surprisingly—the Rh6 cells indeed function as interneurons receiving histaminergic input from Rh1 expressing photoreceptors, and presumably also from R7 and R8 cells. These are very interesting and important findings, because they reveal a similarity with the mammalian system, where melanopsin expressing retinal ganglion cells function both as light receptors and interneurons (receiving signals from rods and cones) in circadian clock resetting. The experiments are well conducted, and involve an impressive generation and collections of mutant combinations. The manuscript is also very well and clearly written and should be understandable for most readers of *Nature Communications*. While most of the conclusions are well supported by the data presented, this is not the case entirely. For example, (unless I missed something) it is not clear if Rh6 expression in the R8 cells really contributes to light resetting from the experiments shown here. Also, the behavioral analysis lacks proper quantification of the data and statistical analysis to allow a clear judgement about synchronization of a certain genotype. This is an important issue, because the ability to synchronize or not (or fast versus slow), is then used to draw conclusions about the network organization.

Major points:

1) the authors claim that the CHO mutants do not entrain to advanced or delayed LD cycles (Fig 1A and ED1, respectively). But in the advance paradigm it seems that the flies slowly advance their activity. In the legend it is argued that this is due to a short free running period, but this can't be true, because the same flies do not show this short period during the delay experiment (ED Figure 1). So this issue must be dealt with and the authors should also tabulate DD periods for all genotypes to back up their claim.

2) The red dots indicating the activity peak are very helpful, but is not clear how they were determined. In fact looking at the CHO mutant in Fig 1A, it seems that the flies are almost constantly active during the 24 hr day (in particular during the entire light phase) and it is not clear how the position of the red dot was determined. Also, since these are average actograms, is it possible to calculate error bars for each of the dots (similar to what was done for ED Fig 4) and do perform statistical analysis to see which genotypes are significantly different? Another possibility that would help determining if entrainment has occurred would be to show daily average histograms for the periods before, during and after the shift of the LD cycle.

3) p3 of results, middle of 2nd last paragraph and ED Fig 3: The authors state the only Rh6-gal4 driven HisCl1 expression rescues resynchronization in CHO flies. While this Rh6 driven rescue is indeed clear, I am not sure about the non-rescue with Rh5-gal4 (ED Fig 3). The flies seem to slowly re-synchronize during the shifted LD, because they show a 24 hr period during the final DD. Again, a period determination during and after entrainment will hopefully help to solve this issue. If the Rh5-gal4 driven HisCl1 expression indeed does not lead to rescue, this would mean that the H-B eyelet does not contribute to this light input pathway, because both Rh5-gal4 and Rh6-gal4 are expressed in the H-B eyelet (e.g. Veleri et al 2007). If true, this would support the results shown in Fig 4B for the so[1] mutant. But here (Fig 4B) again, I find it very difficult to judge if the flies with HisCl1 expression in the 'H-B eyelet only' show rescued behavior or not. It seems the phase advances for about 3 days in the direction of the LD cycle, but then adjusts overall to a later phase compared to the so[1] cry double mutants. Some kind of quantification of the ability to synchronize indeed seems required (see comments above).

4) Results page 4, 2nd paragraph and Fig 4A: The results for the Rh6>HisCl; CHO, ninaE flies are convincing (and quantified) and I therefore can follow the argument that Rh1 in R1-R6 cells is the source for histaminergic input into the Rh6 cells. But in the next sentence the authors claim that that Rh6 cells receive histaminergic input from both Rh1 and Rh6 expressing photoreceptors. Are they suggesting autocrine signaling for the Rh6 expressing R8 cells, based on the ability of these flies to slowly resynchronize to red light LD due to Rh6 function in R8 cells? And I am confused by the data shown in Fig 4A in the actogram next to the ninaE mutant, i.e. the ninaE, Rh6 double mutant. What is the phenotype here? It is not explained in the text (or I missed it) and looking at the actogram it is very difficult to judge what is going on (also for some reasons the red dots are in a different part compared to all the other actograms, why is this?). In any case, to me it looks as if the Rh6>hisCl1 CHO, ninaE Rh6 mutants quickly resynchronize to the shifted red LD cycle, and then they shift again after about 4-5 days. If I understood correctly, these flies should NOT be able to re-synchronize to red LD, because they are lacking the red sensitive rhodopsins Rh1 and Rh6. Please explain.....

Minor comments:

1) Summary: Rh7 expression within clock neurons and retina is controversial. The 2 references cited here suggest that Rh7 is expressed in the LNv clock neurons and not in the retinal photoreceptors (Ni et al 2017, using Rh7 antibody and RT-PCR), whereas the other study suggests an equal distribution of Rh7 expression between retina and brain (using qPCR) (Kistenpfenning et al 2017). Using a reporter line and Rh7 expression the latter study suggests that Rh7 is expressed in the fenestrated layer of the lamina, the R8 cells, the H-B eyelet, and a subset of the DN1 clock neurons. Since at least one study therefore suggests that Rh7 is expressed in the R8 cells, a cell type crucial for HisCl1 function, I think the possibility of Rh7 involvement in some of the processes studied here, should be discussed.

2) It was tested if Ort and HisCl respond to both the norpA-dependent and norpA-independent light transduction pathways. I think for the norpA-independent pathway the experiments conducted are very clear and easy to follow because norpA-mutants were applied. But for the norpA-dependent pathway lower light intensity was used and for most readers it will not be clear why this condition is a test for norpA-dependent synchronization. This needs to be better explained and I suggest to also add a norpA cry mutant control in low light, showing that the flies not synchronize.

3) The above also applies to the use of the red light experiments. A brief explanation for why this condition reveals Rh1 and Rh6 function should be given rather than just citing the relevant reference.

4) It is very helpful that the authors came up with an abbreviation for the HisCl1 Ort cry treble mutant (CHO), but I think it should briefly be stated what it stands for (I think it is Cry, HisCl1, Ort, but I am not sure).

5) (related to Major point 1 above) In ED Fig. 2 several genotypes show short periods during LD-advance and free-run, which is interpreted as lack of synchronization. The authors should perform period analysis separately for the LD advance and DD portion of the experiment to confirm this. Are the short periods also observed in delay experiments? At least for the CHO mutant this seems not to be the case (ED Fig 1) but have other mutant (non-synchronizing) mutants been tested?

6) Results page 2: Sentence starting with the 'The small and large LNvs...', remove 'and large', because it is repeated later in the sentence.

7) Fig 2C: Say in which clock neurons Clk(6-1) gal4 is expressed.

8) p3 of results, end of 2nd last paragraph: That HisCl1 cannot replace Ort function in the optic lobe is shown in Fig 2B, and not Fig 2A as stated.

Reviewer #1 (Remarks to the Author):

This is a terrific contribution. The scale and scope of the genetic experiments is remarkable. These make it possible to better understand CRY-independent entrainment pathways. They distinguish roles for two different histaminergic pathways in the fly eye/brain. They lead to the fascinating model in which (akin to the design for mammalian photoentrainment) a key photoreceptor class (Rh6-expressing) provides entrainment information to central pacemakers by incorporating synaptic inputs from other photoreceptors. The work is novel, well presented and of interest to a broad audience. I have one general concern about the analysis of behavioral synchronization as described below.

My main (and only) difficulty concerns the definitions by which flies were scored for light synchronization. No strict definition or quantitative landmark is applied; instead the authors offer a subjective determination based on the phase of the evening activity. In so doing the authors are in fact well within the traditions of field. However, I recommend in this case a more objective measure could be valuable.

We acknowledge that a more quantitative way of defining synchronization would be better. Based on our entrainment protocol, we now test synchronization as indicated in the Methods section (Extended data): “The statistical analysis of the phase data was done as follows. For each day, the ZT that corresponds to the peak value of evening activity (second half of the day) in LD/RD cycles was determined from the averaged activity data (n flies). These values were used to generate phase plots that are shown next to the corresponding actograms to better visualize the daily phase during the entire experiment. Entrainment was then estimated by comparing the peak ZT values of the five last days of the shifted LD/RD regime to those of the control (Canton S). A genotype was considered entrained when this series of five values was not significantly different from the control one in a one-way ANOVA Dunnett's multiple comparisons test (GraphPad Prism 7 software). Results of the ANOVA are shown in ED Table 2.

The phase data and the results of the statistical test are shown in ED Table 2 for the all actograms shown in the four main figures and in ED figure 3, which contains the Rh4-gal4-driven HisCl1 rescue (see below). In addition, we have added phase plots for all actograms of the four main figures.

Early on in this manuscript, rhythmic behavior is scored either “synchronized” or “not”, as in Figure 1. In Fig 1B all four genotypes are said to show “efficient synchronization” to the light conditions. However, I find that the Rh6[1], ort[1] combination entrains poorly compared to the other three: it appears to need 4 (or maybe even 5) cycles to finally achieve proper register compared to others.

We ask here whether the considered genotypes are able to synchronize or not within the 8 days of the shifted RD cycle to determine whether each of the two histamine receptors receives signals from both Rh1 and Rh6 photoreceptors. It is thus mostly a yes/no question and the answer is yes. However, we definitely agree with the reviewer that all genotypes do not synchronize with the same speed. We now indicate in the text that the Rh6- ort- double mutant synchronizes to RD cycles more slowly than the other double mutants.

Later, the authors also discriminate on the basis of how quickly flies display synchronization: “Control flies phase-shifted within two days but HisCl1 RNAi flies took about five days to synchronize with the new light regime (Fig. 3N). We conclude that in such conditions, HisCl1 is required for fast synchronization with LD cycles.”

Also in Fig 4A, the introduction of an Rh1 mutation is said to slow synchronization. But the basis by which the speed is judged in this case (and not in Figure 1) is unclear.

In these two cases (HisC11 RNAi and Rh1-), and now the new with HisC11- eye clone advanced LD experiment (Fig. 4c), we ask whether the genetic manipulation that is introduced affects entrainment. Since these manipulations do not abolish synchronization (as indicated by the statistical test), we ask whether they slow it down by comparing the phase value of each day between the tested genotype and the control. We now explain this more clearly in the text and provide this comparison of the synchronization kinetics between mutants and controls for the three genotypes in ED Fig 4.

Finally, on p. 5: “In contrast, none of the other Rh-gal4 rescued entrainment (ED Fig. 3), showing that HisC11 acts specifically in Rh6-expressing photoreceptors.” To my eye, Rh4 provided entrainment after 5 days - slow but clear. Was a minimum number of synchronized cycles required? In the absence of a quantitative definition for synchronization, the basis for many conclusions are unclear.

As indicated above, we now more objectively define entrainment by statistically testing the difference between mutants and wild type flies in the last five days of the shifted LD cycle. Based on this test, Rh4 > HisC11 do not synchronize. We cannot exclude that they would eventually synchronize if exposed to the new LD regime for a longer time, but this could also apply to other genotypes as well. We now indicate this in the Methods.

Minor points:

1. *“In contrast to vision, circadian photoreception can thus be independently supported by Ort and HisC11”.*

Agreed, but their equivalence could be a function of a laboratory setting: they may not be completely redundant in nature

We agree and we now address this point in the discussion.

2. *Finally, we addressed the retinal expression of the ort and/or HisC11 genes with a new sensitive in situ hybridization technique.*

Suggest adding a citation to this statement .

We now refer to the Methods section, which includes references.

3. *I found no Methods section or even a reference to a section on-line. Is Methods description limited to the Figure Legends?*

I guess this is a question for the Editors.

The Methods section is in the Extended Data

Reviewer #2 (Remarks to the Author):

This paper, along with a recently published Cell paper on color opponency (Ref 17), shows that there is communication between photoreceptors in the Drosophila retina. In this case, it is to entrain the circadian clock. Thus, this shows that the fly retina, like its mammalian counterpart, contributes to visual processing rather than simply detecting light.

The nature of clock entrainment by light in flies has been a vexing problems for years since it appears

that there is extensive redundancy, with Cryptochrome in clock neurons, photoreceptors in the retina and the eyelet, a remnant of the larval eye that, with its four photoreceptors, also contributes to light entrainment. This means that it takes multiple mutants to dissect the process. To make things worse, a recent report showed that an atypical rhodopsin, Rh7, is also expressed in clock neurons and participates to clock entrainment through a non-classical phototransduction cascade. Strangely though, most recent papers on the clock appear to ignore Rh7, and the current paper does as well. Photoreceptors are activated by light but are histaminergic and therefore, inhibit their targets in response to light. Mammalian photoreceptors do the exact opposite. Since most of the histaminergic neurons are the photoreceptors, it is often assumed that any cell that expresses the Histamine receptor Ort is a target of photoreceptors. However, there is another understudied Histamine receptor, HC11 that was totally ignored until the recent report that shows that it is involved in synapses between R7 and R8 for color opponency. The current paper shows that photoreceptors also talk to one another, but in this case to mediate clock entrainment. The authors show that the Rh6-expressing green photoreceptors receive input from outer photoreceptors as well as from UV-sensitive R7 and through a mysterious way, instruct the clock: this means that R8 serves as an interneuron by collecting information from other photoreceptors. The synapses between photoreceptors are not using Ort but instead, the special histamine receptor HC11 (as for color opponency) to entrain the clock: therefore, this is the second example of local processing in the fly retina, which is highly unexpected and exciting.

The paper describes a large number of manipulations with often very complex genotypes and these indicate that the model is likely to be correct.

However, there is an obvious and critical experiment that is missing that makes me believe that things might not be so simple: deleting or silencing (or activating since Histamine is inhibitory) the Rh6 expressing R8 cells should suppress entrainment and demonstrate its role as an interneuron. The authors must have performed this very simple experiment (which they have done for Rh5-expressing R8!).

We agree that killing Rh6 cells would be a very nice experiment and we had tried to do this before. Unfortunately, we could not obtain an efficient ablation of photoreceptors, in agreement with the information that could obtain from eye labs. We did see many remaining Rh6-positive cells in the Rh6>DTI flies, and driving UAS-hid rpr in Rh6 cells did not do better. We also tried UAS-Kir to silence the Rh6-expressing R8 cells but this did not give any significant synchronization defect. We now mention this in the text. We also had a partial Rh5 cell killing with Rh5-gal4 in the eye, but the eyelet was completely gone (indicated in the text). Likely, the very early expression of Rh5 in the bolwig organ larval photoreceptors makes a difference with the retinal photoreceptors.

We have now replaced the Rh5-killing genotype by a new one where Rh5-cell killing is done in a Rh6>HisC11 rescue background (Fig. 5b). This shows that the Rh6-dependent function of HisC11 does not require the eyelet.

Importantly, we produced flies with HisC11 mutant eyes by mitotic recombination and such flies did show a very pronounced decrease of synchronization speed with an advanced LD cycle and no synchronization with a delayed LD cycle, as shown in Fig 4c and ED Table 2. This strongly reinforces the conclusions of the Rh6-targeted HisC11 RNAi experiments and thus show that the eye-mediated role of HisC11 is critical.

*They also argue that outer photoreceptors (as well as R7) signal to Rh6-expressing R8. However, synapses between outer and R8 photoreceptors have not been reported in the connectome from *Janelia*. The role of R7 is also not tested although it would be a trivial experiment to do so (using sevenless mutants).*

Therefore, showing (simple) processing in the fly retina is an important finding that would reinforce the data published for color vision and would have significant evolutionary implications.

Once the missing experiments are performed (delete Rh6-expressing R8, use sev mutants, eventually show that outer photoreceptors synapse to R8), the paper should be published as a very significant piece of work.

We understand that it would be very nice to describe the synaptic connections that support our behavioral results but we believe that it is out of the scope of this paper.

The role of R7 cells is tested in the *Rh5², Rh6-gal4/ Rh5², UAS-HisC11; CHO, ninaE¹⁷ Rh6¹* genotype (Fig 5c). These flies do not have Rh1, Rh5 and Rh6 and thus only have functional R7 cells to support their synchronization to LD cycles. This shows that R7 input is sufficient for entrainment through HisC11 function in Rh6-expressing R8 cells. However, the ability of *CHO Rh6>HisC11* flies to entrain in red light indicates that R7 input is not required.

Reviewer #3 (Remarks to the Author):

Review for Alejevski et al.

This is a very interesting paper from the Rouyer group about the role of the 2 histamine-gated Chloride channels encoded in the Drosophila genome in synchronizing the circadian clock to light-dark cycles. It is well known that light-resetting in Drosophila involves the blue-light photoreceptor Cryptochrome (Cry) expressed in about 50% of the clock neurons as well as visual receptors in the compound eye. Removing both Cry and histamine—the principal neurotransmitter of the visual system, results in flies that can no longer be synchronized to light. But how the visual system signals to the clock neurons is so far not know. This study supplies a first attempt by resolving how and where the 2 histamine receptors function in light resetting. The authors show that both receptors contribute to light resetting and that the relevant expression of HisC11 occurs in the Rh6 expressing R8 photoreceptor cells, whereas that of Ort is in the optic lobe interneurons. Interestingly, Ort can replace HisC11 function in the R8 cells but not vice versa. The authors then go on to show that—very surprisingly—the Rh6 cells indeed function as interneurons receiving histaminergic input from Rh1 expressing photoreceptors, and presumably also from R7 and R8 cells. These are very interesting and important findings, because they reveal a similarity with the mammalian system, where melanopsin expressing retinal ganglion cells function both as light receptors and interneurons (receiving signals from rods and cones) in circadian clock resetting. The experiments are well conducted, and involve an impressive generation and collections of mutant combinations. The manuscript is also very well and clearly written and should be understandable for most readers of Nature Communications. While most of the conclusions are well supported by the data presented, this is not the case entirely. For example, (unless I missed something) it is not clear if Rh6 expression in the R8 cells really contributes to light resetting from the experiments shown here. Also, the behavioral analysis lacks proper quantification of the data and statistical analysis to allow a clear judgement about synchronization of a certain genotype. This is an important issue, because the ability to synchronize or not (or fast versus slow), is then used to draw conclusions about the network organization.

Major points:

1) the authors claim that the CHO mutants do not entrain to advanced or delayed LD cycles (Fig 1A and ED1, respectively). But in the advance paradigm it seems that the flies slowly advance their activity. In the legend it is argued that this is due to a short free running period, but this can't be true, because the same flies do not show this short period during the delay experiment (ED Figure 1). So

this issue must be dealt with and the authors should also tabulate DD periods for all genotypes to back up their claim.

We now provide the DD periods (and associated powers) for all genotypes of the main figures (and ED Fig. 3) in ED Table 1. The CHO period is indeed slightly short (23h) in the experiment shown in Fig. 1. The same genotype can have a period close to 24h in other experiments, as seen in ED Fig. 1. In other cases, such as HisC11 ort double mutants, the Fig. 1 experiment shows a 23.6h period and, based on the visual inspection of the actogram, a slightly shorter period is observed in ED Fig 1. Small period differences are common in DD experiments with 10-25 flies over 5-7 days and we don't think that any conclusion can be drawn from such variations. Another example disconnecting period and LD shift is now included in fig 4c (bottom): flies with HisC11 mutant eyes show a short period with a delayed LD cycle.

2) The red dots indicating the activity peak are very helpful, but is not clear how they were determined. In fact looking at the CHO mutant in Fig 1A, it seems that the flies are almost constantly active during the 24 hr day (in particular during the entire light phase) and it is not clear how the position of the red dot was determined. Also, since these are average actograms, is it possible to calculate error bars for each of the dots (similar to what was done for ED Fig 4) and do perform statistical analysis to see which genotypes are significantly different? Another possibility that would help determining if entrainment has occurred would be to show daily average histograms for the periods before, during and after the shift of the LD cycle.

The red dots represent the peak of activity for each day and they are determined by visual inspection of daily activity histograms (averaged for n flies). To improve clarity, we have introduced phase plots for all the actograms of main figures, each dot represents the same maximum activity value of the histogram (average for n flies) for the considered day. Because the peak value is determined on the average histogram, we cannot provide an error bar. In ED Fig 4 (top panel), the error bars are determined from the comparison of three different experiments, and for each one the peak value is an average of n flies. We agree that it would be even better to calculate individual peak values and provide error bars for one experiment. However, we could not determine a reliable peak with activity data from a single fly on a single day and thus considered that providing average values was the best way to go.

We could produce histograms averaged for n flies and x days of the shifted LD cycle but the activity peak changes every day and eventually stabilize at a different day in each genotype. As a consequence, it would be very difficult for the reader to extract significant information out these histograms. We thus decided to draw phase plots, which are now shown for all actograms of the main figures (and ED Fig. 3) and use these values to do a statistical test that is shown in ED Table 2.

3) p3 of results, middle of 2nd last paragraph and ED Fig 3: The authors state the only Rh6-gal4 driven HisC11 expression rescues resynchronization in CHO flies. While this Rh6 driven rescue is indeed clear, I am not sure about the non-rescue with Rh5-gal4 (ED Fig 3). The flies seem to slowly re-synchronize during the shifted LD, because they show a 24 hr period during the final DD. Again, a period determination during and after entrainment will hopefully help to solve this issue. If the Rh5-gal4 driven HisC11 expression indeed does not lead to rescue, this would mean that the H-B eyelet does not contribute to this light input pathway, because both Rh5-gal4 and Rh6-gal 4 are expressed in the H-B eyelet (e.g. Veleri et al 2007). If true, this would support the results shown in Fig 4B for the so[1] mutant. But here (Fig 4B) again, I find it very difficult to judge if the flies with HisC11 expression in the 'H-B eyelet only' show rescued behavior or not. It seems the phase advances for about 3 days in the direction of the LD cycle, but then adjusts overall to a later phase

compared to the so[1] cry double mutants. Some kind of quantification of the ability to synchronize indeed seems required (see comments above).

We now provide a DD analysis, a phase plot and a statistical test showing that Rh5>UAS-HisC11 do not behave like synchronized controls in the last five days of the shifted LD cycle. The HisC11 rescue in “eyelet only” has a noisy activity profile, likely due to the anatomical defects that are induced by the so mutation, but the phase plot convincingly show that they do not synchronize. The results thus indicate that HisC11 expression in the eyelet is not sufficient for entrainment. This conclusion is now reinforced by doing the Rh5-killing (thus no eyelet) in a Rh6>HisC11 rescue background (Fig. 5b). We thus conclude that the Rh6-dependent function of HisC11 does not require the eyelet, but we do not conclude that the eyelet does not contribute.

4) Results page 4, 2nd paragraph and Fig 4A: The results for the Rh6>HisC1; CHO, ninaE flies are convincing (and quantified) and I therefore can follow the argument that Rh1 in R1-R6 cells is the source for histaminergic input into the Rh6 cells. But in the next sentence the authors claim that that Rh6 cells receive histaminergic input from both Rh1 and Rh6 expressing photoreceptors. Are they suggesting autocrine signaling for the Rh6 expressing R8 cells, based on the ability of these flies to slowly resynchronize to red light LD due to Rh6 function in R8 cells? And I am confused by the data shown in Fig 4A in the actogram next to the ninaE mutant, i.e. the ninaE, Rh6 double mutant. What is the phenotype here? It is not explained in the text (or I missed it) and looking at the actogram it is very difficult to judge what is going on (also for some reasons the red dots are in a different part compared to all the other actograms, why is this?). In any case, to me it looks as if the Rh6>hisC11 CHO, ninaE Rh6 mutants quickly resynchronize to the shifted red LD cycle, and then they shift again after about 4-5 days. If I understood correctly, these flies should NOT be able to re-synchronize to red LD, because they are lacking the red sensitive rhodopsins Rh1 and Rh6. Please explain.....

The slowed down synchronization of Rh6>HisC11 in the absence of Rh1 (Fig. 4A and ED Fig. 4) supports a contribution of Rh1 to entrainment, but the flies still synchronize, indicating that other histaminergic inputs contribute. Since this experiment is done in red light, the only rhodopsin source that can contribute to this slow but very clear synchronization without Rh1 is Rh6 (Hanai et al. 2008). The next genotype of the panel is Rh6>HisC11 in a Rh1- Rh6- double mutant, which expectedly does not synchronize with red light. The new phase plot and ED table 2 clearly show that these flies do not entrain and we now better explain this in the text.

Minor comments:

1) Summary: Rh7 expression within clock neurons and retina is controversial. The 2 references cited here suggest that Rh7 is expressed in the LNv clock neurons and not in the retinal photoreceptors (Ni et al 2017, using Rh7 antibody and RT-PCR), whereas the other study suggests an equal distribution of Rh7 expression between retina and brain (using qPCR) (Kistenpfenning et al 2017). Using a reporter line and Rh7 expression the latter study suggests that Rh7 is expressed in the fenestrated layer of the lamina, the R8 cells, the H-B eyelet, and a subset of the DNI clock neurons. Since at least one study therefore suggests that Rh7 is expressed in the R8 cells, a cell type crucial for HisC11 function, I think the possibility of Rh7 involvement in some of the processes studied here, should be discussed.

We now mention these important but controversial data in the introduction. However, we don't think that this debate is relevant for the discussion about HisC11 function, which makes R8 cells able to

respond to histamine, in the presence or absence of Rh6. We don't really see how the presence or absence of Rh7(or any other rhodopsin) in R8 cells would affect HisC11 function.

2) *It was tested if Ort and HisCl respond to both the norpA-dependent and norpA-independent light transduction pathways. I think for the norpA-independent pathway the experiments conducted are very clear and easy to follow because norpA-mutants were applied. But for the norpA-dependent pathway lower light intensity was used and for most readers it will not be clear why this condition is a test for norpA-dependent synchronization. This needs to be better explained and I suggest to also add a norpA cry mutant control in low light, showing that the flies not synchronize.*

We now explain this in the text and the *norpA cry* double mutant in low light has been included in ED Fig. 1

3) *The above also applies to the use of the red light experiments. A brief explanation for why this condition reveals Rh1 and Rh6 function should be given rather than just citing the relevant reference. Indicated in the text.*

4) *It is very helpful that the authors came up with an abbreviation for the HisC11 Ort cry treble mutant (CHO), but I think it should briefly be stated what it stands for (I think it is Cry, HisC11, Ort, but I am not sure).*

Done

5) *(related to Major point 1 above) In ED Fig. 2 several genotypes show short periods during LD-advance and free-run, which is interpreted as lack of synchronization. The authors should perform period analysis separately for the LD advance and DD portion of the experiment to confirm this. Are the short periods also observed in delay experiments? At least for the CHO mutant this seems not to be the case (ED Fig 1) but have other mutant (non-synchronizing) mutants been tested?*

As indicated before, we have included a DD period analysis (ED Table 1) for the main figures.

However, we cannot do this for the LD part of the experiment since the lights-ON and lights-OFF startle responses prevent the determination of the "circadian" period from the activity data. It is true that most of the genotypes show a DD period that is slightly shorter than 24h in the protocol that we use but this also happens with delays (although less often), as seen in ED Fig. 1 for the *cry* and *HisC11 ort* mutant genotypes. A clear example is shown in fig 4c (bottom): non-synchronizing flies with HisC11 mutant eyes show a short period with a delayed LD cycle.

Importantly, this is seen not only non-entrained genotypes but also in entrained ones and we thus believe that it is not related to entrainment.

6) *Results page 2: Sentence starting with the 'The small and large LNvs...', remove 'and large', because it is repeated later in the sentence.*

Small and large applies to aMedulla whereas large only applies to Medulla. We added "also" for clarity.

7) *Fig 2C: Say in which clock neurons Clk(6-1) gal4 is expressed.*

Done

8) *p3 of results, end of 2nd last paragraph: That HisC11 cannot replace Ort function in the optic lobe is shown in Fig 2B, and not Fig 2A as stated.*

Corrected.

REVIEWERS' COMMENTS:

Reviewer #1 (Remarks to the Author):

The revised manuscript does an excellent job of addressing my concerns regarding quantification of entrainment. This was a weakness in the original version, but there is now a clear basis for the authors' conclusions. This was a necessary foundation for this work, and having been installed, I have no other other objections to supporting what is a superior study. It is a highly sophisticated genetic study yet it presents the information in a reasonably digestible form. This work presents fundamental information concerning the cellular basis for setting circadian phase, the poorly-described cellular interactions between retinal photoreceptors, and the potentially conserved features of peripheral processing underlying circadian photoentrainment. For all these reasons, I am a strong supporter for a positive decision.

Reviewer #2 (Remarks to the Author):

I was quite positive about the impact of this paper and supported publication pending a small number of experiments that were surprisingly not in the paper.

In particular, I wanted to know whether killing or silencing Rh6 would affect the phenotype, as would be expected from the extensive set of indirect arguments supporting this view.

However, as I had imagined, the authors had performed this experiments without success, for technical reasons (photoreceptors are difficult to kill). They now report that they cannot conclude based on this experiment.

I should have suggested other means of removing R8 (e.g. late senseless mutants)! Considering the huge amount of work presented in this paper and the set of indirect arguments, I would not want to hold the paper any longer as this would delay publication of an important piece of work.

My other concern was the role of R7 and, if the authors did not do the very simple sevenless experiment, they offer convincing arguments that R7 is sufficient but not necessary.

Finally, I would have appreciated a better discussion of why the connectome does not show outer to R8 contacts, but this is not an essential point.

I therefore support publication of the manuscript.

Reviewer #3 (Remarks to the Author):

The revised version of Alejewsky et al is much improved and all my issues with the original submission have been adequately addressed. Nevertheless, I have a few points I wish the authors to address and clarify, which are outlined below.

Major points:

1) Summary: It is true that Rh7 expression has been suggested to occur in the brain within or near some of the clock neurons. But it also seems to be expressed in subsets of the R8 cells and in the HB-eyelet (Kistenpfenning et al), 2 cell types under study here. I am not asking for any additional experiments, but the authors should discuss the possibility that some of the phenotypes observed here might be mediated by Rh7 function. For example the resynchronization of flies lacking CHO flies lacking Rh1, Rh5 and Rh6, but expressing HisCl1 in Rh6 cells could be mediated by Rh7.

2) Interpretation of the results, and the emerging model for the role of the 2 histamine receptors (in particular HisCl1) are quite complex and sometimes hard to follow. In particular for non-fly people they may be hard to follow, also given by the quite enormous complexity of some of the genotypes involved. I would therefore strongly suggest the authors to generate a model figure/cartoon at the end of the paper to help understanding the reader the main message of the paper.

Minor points:

1) Last sentence at the end of the first paragraph on Page 3 (starting with 'In addition...') needs more explanation to be clear for the reader.

2) Fig. 3 f, g: how can R7 and R8 processes be distinguished here? Which parts of the cells are labelled? Please indicate within figure

3) Figure 4a and ED3: It is surprising that only Rh6-gal4 can rescue entrainment. The authors show in Figure 3 that HisCl1 is also expressed in R7 cells and in the HB-eyelet. So, Rh3 or Rh4-gal4 should also rescue if HisCl1 has a function in R7 and Rh5 should also rescue, if HisCl1 has a function in the H-B eyelet (because Rh5-gal4 is expressed in the HB-eyelet, even though the Rh5 protein is not). Lack of HisCl1 function in the H-B eyelet is also shown in Fig 5b. Could the authors please comment on this? What do they think is the function of HisCl1 in R7 and H-B eyelet, if not entrainment?

4) The genotypes on top of panel 4c should be simplified. For the non-Drosophila reader they are incomprehensible. This also applies for some of the other complex genotypes (e.g. Figure 5b). The

authors should come up with a system that helps the reader to easily comprehend which molecules are mutated, rescued, or knocked down, and in which tissue/cells. Perhaps some simple cartoons above the panels would be helpful here. The full genotypes should go into the legend and/or the Methods part. Also, a cartoon explaining the experiment generating HisCl1 mutant eye clones would be helpful for the reader to be able to understand what was done here.

5) The middle and lower panels of ED Figure 4 are not referred to in the text.

REVIEWERS' COMMENTS:

Reviewer #1 (Remarks to the Author):

The revised manuscript does an excellent job of addressing my concerns regarding quantification of entrainment. This was a weakness in the original version, but there is now a clear basis for the authors' conclusions. This was a necessary foundation for this work, and having been installed, I have no other other objections to supporting what is a superior study. It is a highly sophisticated genetic study yet it presents the information in a reasonably digestible form. This work presents fundamental information concerning the cellular basis for setting circadian phase, the poorly-described cellular interactions between retinal photoreceptors, and the potentially conserved features of peripheral processing underlying circadian photoentrainment. For all these reasons, I am a strong supporter for a positive decision.

Reviewer #2 (Remarks to the Author):

I was quite positive about the impact of this paper and supported publication pending a small number of experiments that were surprisingly not in the paper. In particular, I wanted to know whether killing or silencing Rh6 would affect the phenotype, as would be expected from the extensive set of indirect arguments supporting this view. However, as I had imagined, the authors had performed this experiments without success, for technical reasons (photoreceptors are difficult to kill). They now report that they cannot conclude based on this experiment. I should have suggested other means of removing R8 (e.g. late senseless mutants)! Considering the huge amount of work presented in this paper and the set of indirect arguments, I would not want to hold the paper any longer as this would delay publication of an important piece of work. My other concern was the role of R7 and, if the authors did not do the very simple senseless experiment, they offer convincing arguments that R7 is sufficient but not necessary. Finally, I would have appreciated a better discussion of why the connectome does not show outer to R8 contacts, but this is not an essential point. I therefore support publication of the manuscript.

Reviewer #3 (Remarks to the Author):

The revised version of Alejewsky et al is much improved and all my issues with the original submission have been adequately addressed. Nevertheless, I have a few points I wish the authors to address and clarify, which are outlined below.

Major points:

1) Summary: It is true that Rh7 expression has been suggested to occur in the brain within or near some of the clock neurons. But it also seems to be expressed in subsets of the R8 cells and in the HB-eyelet (Kistenpfenning et al), 2 cell types under study here. I am not asking for any additional experiments, but the authors should discuss the possibility that some of the phenotypes observed here might be mediated by Rh7 function. For example the resynchronization of flies lacking CHO flies lacking Rh1, Rh5 and Rh6, but expressing HisCl1 in Rh6 cells could be mediated by Rh7.

The summary now includes Rh7 putative expression in photoreceptors and a sentence about its putative contribution to the HisCl1 pathway has been added to the discussion, with the Kistenpfenning et al. reference.

2) Interpretation of the results, and the emerging model for the role of the 2 histamine receptors (in particular HisCl1) are quite complex and sometimes hard to follow. In particular for non-fly people they may be hard to follow, also given by the quite enormous complexity of some of the genotypes involved. I would therefore strongly suggest the authors to generate a model figure/cartoon at the end of the paper to help understanding the reader the main message of the paper.

We have added a cartoon as new figure 6.

Minor points:

1) Last sentence at the end of the first paragraph on Page 3 (starting with 'In addition...') needs more explanation to be clear for the reader.

The sentence has been rewritten accordingly.

2) Fig. 3 f, g: how can R7 and R8 processes be distinguished here? Which parts of the cells are labelled? Please indicate within figure

Done.

3) Figure 4a and ED3: It is surprising that only Rh6-gal4 can rescue entrainment. The authors show in Figure 3 that HisCl1 is also expressed in R7 cells and in the HB-eyelet. So, Rh3 or Rh4-gal4 should also rescue if HisCl1 has a function in R7 and Rh5 should also rescue, if HisCl1 has a function in the H-B eyelet (because Rh5-gal4 is expressed in the HB-eyelet, even though the Rh5 protein is not). Lack of HisCl1 function in the H-B eyelet is also shown in Fig 5b. Could the authors please comment on this?

We do not know why HisCl1 supports entrainment only in Rh6-expressing PRs. As indicated in the discussion, one possibility is that Rh6 cells only have the downstream circuit that allows to interpret such (likely) inhibitory signals "Since only Rh6-expressing R8 and not the other inner photoreceptors contribute to this circadian photoreception pathway, Rh6 cells might have specific connections with downstream interneurons. Such specificity has been described for color vision where each of the four inner photoreceptor subtypes connects to a different type of TmY interneuron in the Medulla⁵³."

What do they think is the function of HisCl1 in R7 and H-B eyelet, if not entrainment?

As indicated in the introduction, HisCl1 has a key role in color opponency, which is encoded by inhibitory interactions between R7 and R8 cells (Schnaitmann et al. 2018). Its function in the eyelet remains unknown.

4) The genotypes on top of panel 4c should be simplified. For the non-Drosophila reader they are incomprehensible. This also applies for some of the other complex genotypes (e.g. Figure 5b). The authors should come up with a system that helps the reader to easily comprehend which molecules are mutated, rescued, or knocked down, and in which tissue/cells. Perhaps some simple cartoons above the panels would be helpful here. The full genotypes should go into the legend and/or the Methods part. Also, a cartoon explaining the experiment generating HisCl1 mutant eye clones would be helpful for the reader to be able to understand what was done here.

We definitely agree that some of the genotypes are difficult to understand for the non-specialist reader. We have now added a short description of the genotype below each actogram in the main figures, The method for generating mutant clones in the eye is widely used and we cite the original reference (Stowers and Schwartz, 1999) that already includes a descriptive cartoon.

5) The middle and lower panels of ED Figure 4 are not referred to in the text.

Corrected.